# Positional Vertigo in a Child with Hearing Loss

**DOI:** 10.3390/audiolres16010004

**Published:** 2025-12-23

**Authors:** Melissa Blanco-Pareja, Alberto Vieco García, Teresa Perucho, Javier Santos, José Manuel Moreno-Villares, Nicolas Pérez-Fernández

**Affiliations:** 1Department of Otorhinolaryngology, Clínica Universidad de Navarra, 28022 Madrid, Spainjsantosgarr@unav.es (J.S.); 2Department of Pediatrics, Clínica Universidad de Navarra, 28022 Madrid, Spain; avieco@unav.es (A.V.G.); jmorenov@unav.es (J.M.M.-V.); 3Deaprtment of Genetics, Clínica Universidad de Navarra, 28022 Madrid, Spain

**Keywords:** benign paroxysmal positional vertigo, DFNB16, *STRC*, bilateral moderate sensorineural hearing loss

## Abstract

**Background and Clinical Significance**: Vestibular disorders in children are often overlooked, delaying treatment. Early diagnosis of benign paroxysmal positional vertigo (BPPV) allows for targeted maneuvers during acute episodes. Though rare, BPPV can occur in children due to stereocilin gene (*STRC*) deletions or variants, causing hearing loss and vestibular dysfunction. Case Presentation: This study highlights a case of recurrent vertigo linked to a homozygous deletion on chromosome 15 affecting the *STRC* gene.

## 1. Introduction 

The stereocilin gene (*STRC*) is located on chromosome 15q15.3. The protein is expressed on the surface of cells, particularly in the stereocilia of inner ear hair cells. Variants affecting this gene are associated with autosomal recessive non-syndromic hearing loss, initially named “Deafness, Autosomal Recessive 16” (DFNB16). This condition may also occur with infertility when large contiguous gene deletions at 15q15.3 remove both *STRC* and “Cation Channel, Sperm-associates 2” (*CATSPER2*) genes. Hearing loss in affected individuals is typically mild to moderate, congenital, bilateral, and symmetric, averaging 40–50 dB at diagnosis [1]. Additionally, these individuals may have an increased risk of recurrent benign paroxysmal positional vertigo (BPPV), with one study indicating that 39% of 64 individuals developed BPPV by a median age of 13 years [2].

Benign paroxysmal positional vertigo (BPPV) is an uncommon or misdiagnosed condition in childhood, most frequently occurring in females. In almost 80% of cases, it is considered secondary to head trauma or other general or inner ear conditions or treatments. The main differential diagnosis is Recurrent Vertigo of Childhood (RVC). BPPV can be diagnosed with proper positioning tests during a comprehensive otoneurological examination and treated with appropriate particle repositioning maneuvers, which have a recurrence rate of 14.5% [3].

Early-onset BPPV may have a genetic component due to abnormal structural development of the utricle or semicircular canals [4]. It is noteworthy if there is a familial history of BPPV but must also be considered in patients with bilateral hearing loss, as reported in this case, which prompted us to pursue a genetic diagnosis.

## 2. Case Presentation

The girl in this study was first evaluated in the pediatric and ENT departments at the age of 7 for a recent onset episode of vertigo. Over the preceding two months, the vertigo had fluctuated in severity, leading to five visits to the emergency department for the Epley maneuver. The clinical history was consistent with positional vertigo. During positional testing, the intensity of the vertigo was so severe that it hindered an adequate clinical examination and positioning treatments. Upon evaluation during an apparently quiescent phase, ocular motility was normal, there was no spontaneous nystagmus, and the video head impulse test (vHIT) revealed normal gain and no refixation saccades in all six semicircular canals. The Dix–Hallpike test was negative on the right side but positive when the head was positioned hanging to the left. This elicited intense vertigo and nystagmus after a latency of approximately 5 seconds. The nystagmus was intense, up-beating, and torsional, with the upper pole of the eyes beating to the left, and spontaneously resolved within 15 seconds. On returning the patient to an upright position, vertigo reappeared, and the nystagmus reversed direction. Treatment was initiated with an Epley maneuver, during which an orthotropic nystagmus was observed in the second position. Two weeks later, the patient reported significant improvement, although the Dix–Hallpike test remained mildly positive. A second Epley maneuver was performed, and two weeks later, the episode was considered resolved, as symptoms had disappeared, and clinical examination was negative.

Her previous medical history showed episodes of acute otitis media. The first occurred at the age of 3 and was accompanied by postural instability, prompting a differential diagnosis with cerebellitis and polyradiculitis. A brain MRI and electroneurography were performed, both of which were normal. Difficulties with balance persisted until a first clear episode of vertigo, still at the age of 3, occurring shortly after a second episode of otitis. Vertigo episodes recurred periodically over the next three years: by description, some were positional but others were spontaneous. A CT scan, MRI of the temporal bone, and an electroencephalogram were conducted, but no specific pathology was detected. A possible migraine equivalent was considered. During the second episode of otitis at the age of 4, moderate sensorineural hearing loss was detected, requiring the use of hearing aids. Shortly after, motor and phonic orofacial tics appeared, leading to an evaluation that ruled out PANDAS syndrome.

On follow-up after our first visit, she underwent occupational therapy to support the development of coordination, balance, proprioception, and postural tolerance, as vertigo episodes continued and were classified as BPPV, responding to the Epley maneuver. Around the age of 10, she began experiencing oppressive bitemporal headaches with a kinetic component and dizziness. In the context of relational and academic difficulties, she was diagnosed with high intellectual ability with suspected Asperger’s disorder. She was referred to psychiatry, where autism spectrum disorder was ruled out, but she was diagnosed with social anxiety disorder and generalized anxiety disorder, for which she began psychotherapy. Additionally, she was diagnosed with central precocious puberty, which was monitored by pediatric endocrinology.

Family clinical history related to vestibular and auditory problems was recorded up to four generations, without being significant, since only one paternal grandfather needed the use of hearing aids at an advanced age. There has also been no evidence of consanguinity in the family.

After the first Epley maneuver and during the following 4 years, she was treated for BPPV on six occasions, all of them because of posterior semicircular canalithiasis in her left ear, except for once in the right ear. Hearing levels were monitored yearly during this period. Although the patient denied changes in hearing, minor fluctuations in hearing thresholds (<10 dB) were observed at each follow-up hearing test. Hearing thresholds measured in 2023 showed a clinically significant improvement at 500–1000 Hz in the right ear and at 1000 Hz in the left ear. Hearing thresholds measured in 2024 showed a clinically significant worsening at 500–1000 Hz in the right ear and at 250 and 1000 Hz in the left ear. Hearing loss configuration remained symmetric with a sloping configuration from mild to moderate sensorineural hearing loss throughout all hearing evaluations. Maximum word recognition was 100% at each hearing test. Distortion product otoacoustic emissions (DPOAEs) were absent at 1000–8000 Hz in both ears. An auditory brainstem response (ABR) test was completed using a click stimulus. Waves I–V were registered with normal morphology and at expected latencies, which is compatible with normal neural synchrony. The threshold for wave V for a click stimulus was found at 50 dB nHL in both ears. Vestibular testing showed normal vHIT, vestibular evoked myogenic potentials (VEMPs), both ocular and cervical, were normal in the amplitude and latency of the different waves, and the interaural asymmetry ration was 1% for oVEMP and 7% for cVEMP.

Given the recurrent episodes of benign paroxysmal positional vertigo and the history of moderate bilateral sensorineural hearing loss, a genetic study was decided upon. Nucleic acids extracted from peripheral blood anticoagulated with EDTA were analyzed, and next-generation sequencing (NGS) was performed on a panel of 179 genes associated with hereditary hearing loss, which detects copy number variations (CNVs), deletions or duplications (indels), and variations in the nucleotide sequence (SNVs). This analysis revealed a potential homozygous deletion on chromosome 15, including the STRC gene. To confirm the clinical relevance of this finding, genetic testing using the multiplex ligation-dependent probe amplification technique (MLPA) was performed on the parents, revealing a heterozygous deletion of the *CKMT1B, STRC*, and *CATSPER2* genes in the father (Figure 1) and a heterozygous deletion of the *CKMT1B* and *STRC* genes in the mother. Both parents were carriers of a heterozygous deletion without clinical manifestations, which they transmitted to their daughter with a 25% probability. The patient has inherited deletions of different sizes from each of her parents, resulting in a homozygous loss of gene *STRC* (Figure 2). Therefore, the diagnosis is non-syndromic sensorineural hearing loss, mild to moderate (*DFNB16*) in the patient.

## 3. Discussion

The clinical manifestations and examination findings are closely correlated with the defective gene identified. Autosomal recessive sensorineural hearing loss associated with the *STRC* gene is the second most common cause of hereditary sensorineural hearing loss after *GJB2* mutations, accounting for approximately 2–8% of cases depending on the population studied [5,6]. *STRC*-related hearing loss characteristically presents with bilateral, mild-to-moderate sensorineural hearing loss affecting primarily mid-to-high frequencies (2–8 kHz), with typical onset in childhood or early adulthood, and demonstrates a relatively stable or slowly progressive course [5]. The genotype–phenotype correlation is particularly strong, as complete *STRC* deletions or compound heterozygous mutations consistently produce this distinctive audiological profile, distinguishing it from the more severe, congenital profound hearing loss associated with *GJB2* mutations or the fluctuating hearing loss with vestibular symptoms seen in *SLC26A4*-related disorders [6,7].

Studies on the cochlear epithelium of stereocilin knockout mice (*Strc−/−*) show a less organized alignment of stereocilia tips compared to *Strc+/+* mice. In the knockout strain, tip links at the outer hair cells (OHCs) appear normal; however, there is a striking absence of top connectors [8] or lateral links between adjacent stereocilia. This may explain the loss of cochlear waveform distortion in the early postnatal days, a period when hearing sensitivity and frequency tuning are still nearly normal. OHC dysfunction accounts in our case for the absence of DPOAE as previously reported. Additionally, studies have demonstrated in one series a progressive worsening of hearing loss, though no significant difference has been observed between patients with homozygous *STRC* deletions and those with compound heterozygous mutations [9]. Because of this, it is recommended to treat patients with hearing aids and to monitor their hearing over time. If hearing loss worsens and no longer benefits from hearing aids, cochlear implantation should be considered.

Scanning immunoelectron microscopy reveals that stereocilin forms a ring around the tips of the tallest row of outer hair cell stereocilia, suggesting its role in contact between cilia [10] and with the tectorial membrane [11]. A similar function is observed in the more complex structure of the maculae. Stereocilin’s glycosyl-phosphatidylinositol anchoring likely attaches otoconial membranes to inner ear cells, similar to its connection with the tectorial membrane [12]. There are two other mechanisms for anchoring: long columnar filaments that anchor the apical surface of supporting cells to the gelatinous layer [13] and the glycocalyx that facilitates connections between the sensory hair bundle and the otolithic membrane or cupula [14]. The absence of stereocilin affects only one of these anchoring methods, maintaining the anatomical continuity of the otolithic membrane. This, along with the less prominent findings in the semicircular canals, explains why *Strc−/−* mice lacking stereocilin do not exhibit significant vestibular dysfunction. This is well exemplified in our patient, who did not show any abnormal vestibulo-ocular (angular or lineal) or vestibulo-spinal reflex.

*STRC* pathogenic variants account for 4.08% of *GJB2*-negative congenital hearing loss cases and represent 14.36% of mild-to-moderate hearing loss patients, with biallelic deletions comprising approximately 70% of all *STRC* pathogenic alleles [15]. At the molecular level, the absence of top connectors between stereocilia and the tectorial membrane impairs cochlear amplification in the outer hair cells [16]. The *STRC–CATSPER2* contiguous deletion at 15q15.3 occurs in approximately 36–77% of *STRC* cases and confers risk for deafness–infertility syndrome in males, highlighting the importance of comprehensive genetic counseling [15,17].

Next-generation sequencing with copy number variation detection has significantly improved diagnostic yield to 38–40% in hearing loss cohorts, revealing novel *STRC* variants and enabling early identification for appropriate audiological management. Comprehensive genetic testing strategies, including combined cytomegalovirus screening and multi-gene panels in carrier screening programs, have enhanced the clinical diagnosis and medical management of pediatric patients with isolated hearing loss [18]. Recent advances in dual-vector gene therapy have demonstrated restoration of cochlear amplification and auditory sensitivity in DFNB16 mouse models, offering promising therapeutic avenues for patients with *STRC*-related hearing loss [16].

Vestibular symptoms in all episodes have been consistent with BPPV of peripheral etiology, and repeated Epley maneuvers PRMs have effectively resolved them; given prior evidence of efficacy, we have performed the Epley maneuver. No complications were noted, including migration of otoconia. However, relapses occur frequently but are progressively less severe. Increased understanding of the condition and its recovery has reduced anxiety with each new episode. Based on canalithiasis physiopathology, we believe each episode stems from freely floating otoconia in the vestibule. There is no prophylactic treatment, but addressing vitamin D deficiency, a key risk factor for BPPV recurrence, can help mitigate it [19].

## 4. Conclusions

Recurrent BPPV accompanied by sensorineural hearing loss in a child is an uncommon occurrence. When no definitive cause is identified following thorough clinical evaluation, it is essential to proceed with comprehensive genetic diagnosis.

## Figures and Tables

**Figure 1 audiolres-16-00004-f001:**
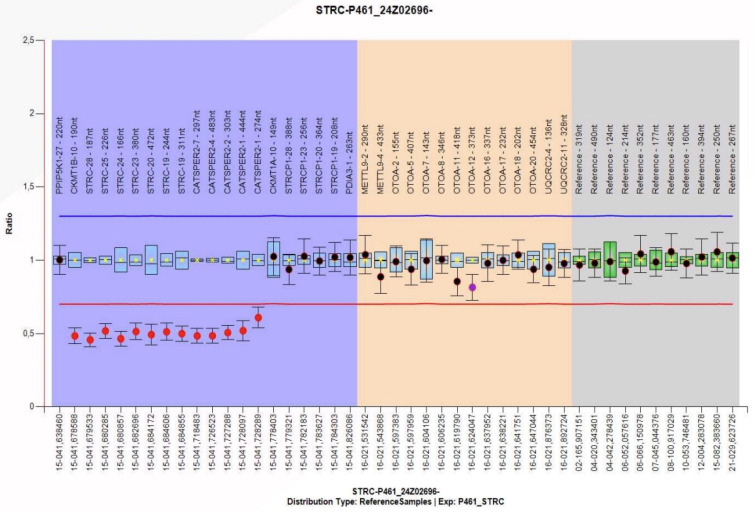
Result of multiplex ligation-dependent probe amplification technique (MLPA) showing deletion of adjacent genes CKMT1B, STRC, and CATSPER2 in the father.

**Figure 2 audiolres-16-00004-f002:**
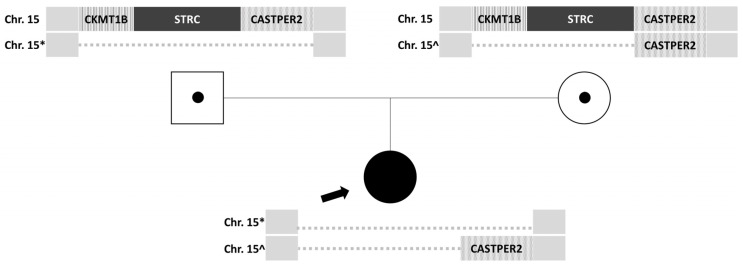
Schematic representation of the inheritance of the homozygous deletion in the proband (filled in, with an arrow) from heterozygous parents with different genotypes (carriers, with dot).

## Data Availability

The original data presented in this study are available on reasonable request from the corresponding author. The data are not publicly available due to privacy concerns.

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
