# Peer review of "Positional Vertigo in a Child with Hearing Loss"

_audiolres, 2025, doi:10.3390/audiolres16010004_

Round 1

Reviewer 1 Report

Comments and Suggestions for Authors

The paper makes a good contribution to the literature.The paper is clear and easy to read. The section on genetic aspects is exhaustive, and the authors' explanation for the recurrence of BPPV appears equally convincing

Author Response

The paper makes a good contribution to the literature.The paper is clear and easy to read. The section on genetic aspects is exhaustive, and the authors' explanation for the recurrence of BPPV appears equally convincing

Thanks a lot!

Reviewer 2 Report

Comments and Suggestions for Authors

Dear authors, the article is interesting and concerns an important topic  but  I  have a few suggestions. .

Abstract

Please, be assured that all the acronyms are explained

Introduction

Additionally, these individuals may have an increased risk of recurrent benign paroxysmal positional vertigo (BPPV), with  one study indicating that 39% of 64 individuals developed BPPV by a median age of 13 31 years (1, 2 ). You wrote one study but citations are 2..

Do  you think it could be useful to better introduce BVVP?

Discussion

The clinical manifestations and examination findings are closely correlated with the defective gene identified -

Can you better explaine?

Autosomal recessive sensorineural hearing loss associated with the STRC gene is the second most common cause of hereditary sensorineural hearing loss  (5). Can you  go into details in this concept?

In my opinion the discussion should better address why the authors had the suspect  of a genetic origin of the symptoms and take in consideration other clinical pictures in differential diagnosis .

 It would also important to  add discussion about maneuvers used.

Author Response

Thanks for the commnets

In th new version of the paper the responses are shown.

Reviewer 3 Report

Comments and Suggestions for Authors

This article reports a rare case of benign paroxysmal positional vertigo in children associated with homozygous deletion of the STRC gene. The article is well written, with complete case information and detailed descriptions of the diagnosis and treatment process. The discussion section combines basic research with clinical phenotypes. However, there are still some issues in the article that need improvement.

  1. Figure 2. The symbols of the proband and parents are both filled with white, which is incorrect. In a family chart, symbols filled with white generally refer to individuals who are not ill or have no  pathological phenotype. The proband should be clearly indicated by an arrow and her symbol should be filled (e.g., black) to denote an affected individual. Please check the figures to avoid similar errors.
  2. The report excellently details the genetic findings in the proband and her parents, confirming the autosomal recessive inheritance pattern of the STRC deletion. However, the clinical family history is not explicitly described beyond stating that the parents are asymptomatic carriers. To strengthen the case and provide a more comprehensive clinical picture, it would be valuable to include a statement regarding the extended family history.
  3. The examination results such as auditory test, CT or MRI need to be provided in Figure.
  4. The relevant and necessary research background has not been fully described and cited. What is the frequency of known STRC gene pathogenic variants among patients with congenital deafness? What pathological changes can Strc gene variantscause in the inner ear of mice or humans? Strc defect cochlea lacked top connectors and had disorganized hair bundles, consistent with loss of contact with the tectorial membrane. However, the author did not provide these necessary information.
  5. When you are supplementing your research background, the following articles are recommending for reference. 1. Dual-vector gene therapy restores cochlear amplification and auditory sensitivity in a mouse model of DFNB16 hearing loss; 2. Next-Generation Sequencing of Chinese Children with Congenital Hearing Loss Reveals Rare and Novel Variants in Known and Candidate Genes; 3. Analysis of the Results of Cytomegalovirus Testing Combined with Genetic Testing in Children with Congenital Hearing Loss; 4. Hereditary deafness carrier screening in 9,993 Chinese individuals; 5. Comprehensive genetic testing improves the clinical diagnosis and medical management of pediatric patients with isolated hearing loss; 6. Frequency of the STRC-CATSPER2 deletion in STRC-associated hearing loss patients.
  6. There are many small errors in the article.

-Line 101, “remined” should be “remained”.

-Line 30, “benign paroxysmal positional vertigo (BPPV)”. The full name and abbreviation have been provided here, while at Line 33, “Benign paroxysmal positional vertigo (BPPV)” reappear.

-Line 103, “oVEMP and 7% for cVEMP”. Abbreviations need to be given their full names when they first appear.

-Line 158, “PRMs”. The full name of this abbreviation should be provided here.

Other typographical errors include underline and bold font, please check carefully.

  1. -Line 162, “There is no prophylactic treatment, but addressing vitamin D deficiency, a key risk factor for BPPV recurrence, can help mitigate it.”Please provide references.

Author Response

Thanks for the comments 

Modifications are shown in red in the text as in figure 2

Reviewer 4 Report

Comments and Suggestions for Authors

Congratulations to the Authors that present a very interesting case report dealing with a rare but important clinical situation. We know that hearing loss in children is often without a certain diagnosis and that vertigo in children is not an easy matter as well. The association among bilateral sensorineural hearing loss, positional recurrent episodes of vertigo and genetic disease leads to this uncommon clinical entity. All parts of this report are clearly and deeply presented. It was a pleasure to read it! No doubt that the readers of this Journal will appreciate it!    

Author Response

Congratulations to the Authors that present a very interesting case report dealing with a rare but important clinical situation. We know that hearing loss in children is often without a certain diagnosis and that vertigo in children is not an easy matter as well. The association among bilateral sensorineural hearing loss, positional recurrent episodes of vertigo and genetic disease leads to this uncommon clinical entity. All parts of this report are clearly and deeply presented. It was a pleasure to read it! No doubt that the readers of this Journal will appreciate it!    

Thanks a lot!

Round 2

Reviewer 3 Report

Comments and Suggestions for Authors I recommend accepting it in its current form.